# Seroepidemiology and the Molecular Detection of Animal Brucellosis in Punjab, Pakistan

**DOI:** 10.3390/microorganisms7100449

**Published:** 2019-10-13

**Authors:** Usama Saeed, Shahzad Ali, Tahir Mahmood Khan, Hosny El-Adawy, Falk Melzer, Aman Ullah Khan, Anam Iftikhar, Heinrich Neubauer

**Affiliations:** 1Wildlife Epidemiology and Molecular Microbiology Laboratory (One Health Research Group), Discipline of Zoology, Department of Wildlife & Ecology, University of Veterinary and Animal Sciences, Lahore, Ravi Campus, Pattoki 55300, Pakistan; usamasaeeduvas@gmail.com; 2Institute of Pharmaceutical Science, University of Veterinary and Animal Sciences, Lahore 54000, Pakistan; tahir.khan@uvas.edu.pk; 3Friedrich-Loeffler-Institute, Institute of Bacterial Infections and Zoonoses, Naumburger Str. 96a, 07743 Jena, Germany; Hosny.ElAdawy@fli.de (H.E.-A.); Falk.Melzer@fli.de (F.M.); amanullah.khan@uvas.edu.pk (A.U.K.); Heinrich.Neubauer@fli.de (H.N.); 4Faculty Medicine of Veterinary, Kafrelsheikh University, Kafr El-Sheikh 33511, Egypt; 5Department of Pathobiology, College of Veterinary and Animal Sciences, Jhang 35200, Pakistan; 6Department of Biological Sciences, University of Veterinary and Animal Sciences, Lahore, Ravi Campus, Pattoki 55300, Pakistan; anam.iftikhar@uvas.edu.pk

**Keywords:** brucellosis, Pakistan, risk factors, seroprevalence, PCR

## Abstract

Brucellosis is an infectious disease caused by bacteria of the genus *Brucella* (*B.*), affecting both animals and humans, causing severe economic loses and severe illness, respectively. The objective of the present study was to determine the seroprevalence and the risk factors associated with caprine, ovine, and bovine brucellosis in selected districts of Punjab, Pakistan. A total of 1083 blood samples were randomly collected from animals (goats = 440, sheep = 203, cows = 206, and buffaloes = 234). Questionnaires were used to collect data on risk factors associated with brucellosis on the sampling day. All samples were initially screened for anti-*Brucella* antibodies using the rose bengal plate test (RBPT). The seropositive serum samples were confirmed by a quantitative real-time polymerase chain reaction (PCR) assay for the detection of the *Brucella* genus- and *Brucella* species-specific DNA (*B. abortus* and *B. melitensis*). Univariant and binary logistic regression were used to identify important risk factors of brucellosis. Anti-*Brucella* antibodies and DNA were detected in 35 (3.23%) serum samples. Thirty-four (97.1%) DNA samples were confirmed as *B. melitensis* by qRT-PCR. Abortion history and natural mating were found to be potential risk factors. *Brucella melitensis* was identified as the causative agent of caprine, ovine, and bovine brucellosis in the selected districts of Punjab, Pakistan. Diseased animals may act as a source of infection for other animals. The elimination of positive seroreactors, development of control strategies for brucellosis, and education programs regarding the control of zoonotic disease are highly needed in developing countries like Pakistan.

## 1. Introduction

Brucellosis is one of the most devastating zoonotic diseases that has spread all over the world. It affects a wide range of animal species and humans [1,2,3,4]. Brucellosis is considered the most critical zoonotic disease after rabies and tuberculosis. The main symptoms of brucellosis in animals are abortion, reduction of milk production, hygroma, neonatal mortality, infertility, epididymitis, and orchitis. The typical symptoms of brucellosis in males are orchitis and epididymitis, and abortion is typically the first recognized clinical sign in pregnant females [5]. Risk factors associated with brucellosis can be categorized into management, animal, and environmental factors. The screening of new arrivals, hygiene, vaccination, size of herd, breeding practices, and production system are management risk factors. Animal risk factors include age, breed, sex, abortion history, and milking method. The agro-ecological location of animals is categorized as an environmental factor [6,7].

Most of the animals which have recovered from brucellosis spontaneously might be shedding the bacteria in urine, milk, and vaginal secretion. When animals are slaughtered, *Brucella* species are also shed outside [8,9]. Brucellosis may lead to serious economic losses through the death of young stock, still birth, or abortion, and efforts for improved breeding are hindered [10]. Livestock provides food, skins, fibers, manure (fertilizer or fuel), and draught power. In developing countries, livestock is the basis of livelihood for about 95% of the rural population [11].

There are several draw backs to diagnose brucellosis using isolation and culture. *Brucella* shows slow growth, which may delay diagnosis for up to 7 days or more. The sensitivity of the culture is also low (50 to 90%), depending on the given species, stage of disease, culture media, quantity of bacteria, and culture technique used. Here, culturing is not economical and is time consuming. Thus, the most important diagnostic screening technique and confirmation method are serological methods [12]. However, no species diagnostic is feasible. The *Brucella* antigen shows cross reactions with some other bacteria species such as, *Vibrio cholera*, *Bordetella bronchiseptica*, *Yersinia enterocolitica*, and the *Salmonella* species [13]. Molecular methods, like polymerase chain reaction (PCR), have proven to be fast, specific, reliable, and safe methods. The most important advantage of RT-PCR is its analytical sensitivity, as can detect the few bacteria present in a sample [4].

In small ruminants, *B. melitensis* is typically the causative agent of brucellosis. However, the most frequent causative agent of bovine brucellosis is *B. abortus*. Infrequently *B. suis* or *B. melitensis* can cause brucellosis in bovines if kept in close contact to pigs, goats, or sheep [14]. Only few studies are available relating to animal brucellosis in Pakistan. The overall seroprevalence of brucellosis reported for Pakistan was 3.25–4.4% in livestock. Based on serum agglutination testing, the seroprevalence was 5.06% and 5.49% in cattle and buffaloes, respectively. The seroprevalence of brucellosis in goats and sheep were found to be 1.94% and 1.47%, respectively [15]. Keeping in view the economic importance of animal brucellosis, the present study was designed to determine the seroprevalence and risk factors associated with caprine, ovine, and bovine brucellosis in the selected districts of Punjab, Pakistan.

## 2. Materials and Methods

### 2.1. Sampling Sites

The study was conducted in Kasur, Okara, Lahore, and Faisalabad in the Punjab province of Pakistan (Figure 1). The Kasur District (31.1165° N, 74.4494° E) is south of Lahore and borders India along the Ganda Singh border. Here, important livestocks are cattle (Sahiwal), buffalo (Nili Ravi), goats (Beetal, Teddy), and sheep (Lohi, Kajli). According to the 9211 Virtual Governance System, the livestock of this district includes large animals (1,060,994), poultry (243,141), and small ruminants (319,048). Different infectious or metabolic diseases are reported from this area, including ketosis, milk fever, foot and mouth disease, hemorrhagic septicemia, etc. Vaccination for these diseases is given to animals [16]. The use of animals for research purposes was performed according to the Institutional Ethical Review Committee for Animal Care and Use, the University of Veterinary and Animal Sciences, Lahore, Pakistan via approval number DR/750. 

The name of the Okara city (30.8090° N, 73.4508° E) is derived from the name of the tree “Okaan”. Here, the livestock population and production is very high. The city is located by the river Ravi. Okara is also famous for its cotton mills and various military dairy farms, and for the cheese of these farms. Important livestock breeds of the Okara district are cattle (Sahiwal, Cholistani), buffalo (Nili Ravi), goats (Beetal, Teddy), and sheep (Kajli). According to the 9211 Virtual Governance System, the livestock of this district includes poultry (296,097), large animals (1,312,106), and small ruminants (474,062). The prevalent diseases of this area related to livestock are milk fever, mastitis, and foot and mouth disease. Vaccination for these diseases is given to animals [16].

The capital of the province Punjab is Lahore (31.5204° N, 74.1350° E), which is the second densest city of Pakistan. It is located near the Indian border Wagah and river Ravi. In Lahore, famous livestock breeds are cattle (Sahiwal) and buffalo (Nili Ravi). According to the 9211 Virtual Governance System, the livestock of this district includes small ruminants (141,651), poultry (122,379), and large animals (604,565). Different initiatives have been taken to improve the livestock sector in this area, such as reducing calf and feed lot fattening, disease control compartment, milk awareness campaigns, poultry unit distribution, distribution of heifer cows/buffaloes, sheep and goats. The prevalent diseases of this area related to livestock are milk fever, mastitis, the herpes simplex virus, bovine ephemeral fever, and foot and mouth disease. Here, vaccinations are applied to animals [16].

Faisalabad (301.4504° N, 73.1350° E) is the third largest city of the province Punjab. In District Faisalabad, important livestock breeds are goats (Beetal), buffalo (Nili Ravi), cattle (Desi, cross breed), and sheep (Lohi). According to the 9211 Virtual Governance System, the livestock of this area includes goats (528,203), cattle (534,499), sheep (87,691), buffalo (999,087), rural poultry (285,492), and pet/captive birds (125,250). Here, the prevalent diseases related to livestock include black quarter red metabolic water, milk fever, foot and mouth disease, and herpes simplex virus infection. Here, vaccinations are applied to animals [16].

### 2.2. Epidemiological Data Collection

The demographic (district and tehsil) and epidemiological data related to risk factors of brucellosis (i.e., species, gender, breed, animal’s age, herd size, abortion history, have their own sire, cleaning up the coral, body condition and stock replacement) were collected using a questionnaire.

### 2.3. Blood Sampling

A total of 1083 blood samples from goats (*n* = 440), sheep (*n* = 203), buffaloes (*n* = 234) and cows (*n* = 206) were collected randomly from the districts Kasur, Okara, Lahore, and Faisalabad. Blood samples (4 mL) were collected into non-EDTA tubes from each animal aseptically from jugular venipuncture with disposable needle. These samples were immediately stored at 4 °C in ice boxes and transported to the Epidemiology and Microbiology Laboratory (One Health Research Group), Department of Wildlife and Ecology, University of Veterinary and Animal Sciences, Lahore, Ravi Campus, Pattoki.

### 2.4. Serum Separation

After the transportation of blood samples to the laboratory, they were kept upright at 4 °C for a maximum of 24 h. All tubes with blood were centrifuged at 5000 rpm for 5 min for serum separation. After centrifugation, the supernatants were collected in sterile Eppendorf tubes (1.5 mL) by pipettes and stored at −20 °C for further analysis.

### 2.5. Rose Bengal Plate Test (RBPT)

Serum samples were analyzed with the rose bengal test (RBT) antigen for the detection *Brucella* antibodies (IDvet France). Brucellosis positive and negative control sera were provided by the Friedrich Loeffler Institute, Jena, Germany, for checking the RBT antigen before analyzing the serum samples. For the serological analysis of the serum samples, equal volumes (30µL) of serum samples were mixed with the RBT antigen. If agglutination was observed (after 4 min), samples were considered positive, otherwise they were considered negative for brucellosis [16].

### 2.6. DNA Extraction

A total of 35 seropositive samples were used to extract DNA. A high purity PCR template preparation kit (Roche Diagnostic, Mannheim, Germany) was used for the extraction of DNA according to manufacturer’s instructions. A ND-1000 UV visible spectrophotometer (Nano-Drop technologies, Wilmington, DE) was used for checking the purity of DNA and its concentration. Then, the DNA samples were stored at −20 °C for further analysis.

### 2.7. Quantitative Real-Time (RT) PCR

Serum samples positive in serology were further confirmed for the presence of *Brucella* DNA using real-time PCR. Two species specific (*B. abortus* and *B. melitensis*) and one *Brucella* genus specific qRT-PCR assays were used [17]. The detailed procedure has been previously described [18]. The primers and probes used here were supplied by TIB MOLBIOL (Berlin, Germany) (Table 1).

The PCR reaction and analysis were performed using the Mx3000P Thermocycler (Stratagene, Canada). The samples scored positive by the instrument were additionally confirmed by visual inspection of the graphical plots, which show cycle numbers versus fluorescence values [19]. Reference strains of *B. abortus* S-99 (ATCC 23448) and *B. melitensis* 16M (ATCC 23456) were provided by the National Reference Laboratory (NRL) as a positive control. A non-*Brucella* gram-negative strain was used in the present study to evaluate the specificity of the primers and real-time PCR reaction, namely, *E. coli* (ATCC 10538). A sample with a fluorescence signal 30 times greater than the mean standard deviation in all wells over cycles 2 through 10 was considered as a positive result, whereas a sample yielding a fluorescence signal less than this threshold value was considered negative. Cycle threshold values below 38 cycles were interpreted as positive. The threshold was set automatically by the instrument. The samples scored positive by the instrument were additionally confirmed by visual inspection of the graphical plots, which show cycle numbers versus fluorescence values.

### 2.8. Statistical Analysis

The Statistical Package for Social Sciences (SPSS), version 21.0, software package was used to perform the statistical analysis of all collected data. The chi square test was used to assess the association between each risk factor and the outcome variable. To estimate the prevalence, binomial logistic regression was ran to calculate the confidence interval (CI). To determine the association between risk factors and the prevalence of brucellosis, odds ratios, along with a 95% CI, were calculated. Binary logistic regression was used to estimate the association between the seropositive and explanatory variables. *P*-values ≤ 0.05 were considered statistically significant for all analyses here.

## 3. Results

Thirty-five (3.2%) serum samples were positive for *Brucella* antibodies using the RBPT. The seroprevalence of *Brucella* antibodies was higher (11.3%) in the Faisalabad District as compared to the Okara (4.1%), Lahore (2.0%), and Kasur districts (1.3%) (Table 2). The seroprevalence of brucellosis significantly (*p* < 0.05) varied from one tehsil to another, where it was highest in Chunian (16.7%) and lowest in Pattoki (1%) (Table 2).

The prevalence of *Brucella* antibodies based on the animal species level were higher in buffaloes (5.1%) than cows (3.8%), sheep (3.4%), and goats (1.8%) (Table 3). The seroprevalence of brucellosis was much higher in males (7.4%) as compared to females (2.5%). The seroprevalence of different breeds of buffaloes and cows was 5.1% in Nili Ravi and 3.8% in Rojhan, respectively. In different breeds of sheep and goats, it was 3.4%, 1.9%, and 1.8% in the Kajli, Teddy, and Beetle breeds, respectively. However, the prevalence of brucellosis was not statistically significant here (*p* > 0.05). In term of age groups, a positive sample was not found in offspring, but Brucella antibodies were found in 3.3% of adults. Here, age group was not statistically significant (*p* > 0.05).

Overall, 19 (12.75%) herds were found to be positive, among these, six were goat herds, five were buffalo herds, four were cow herds, and four were sheep herds. The seroprevalence of *Brucella* antibodies was higher in large herds (26.4%) than in smaller herds (5.2%). The association of infection with increasing herd size was statistically significant (*p* < 0.05).

A higher seroprevalence (9.0%) was reported in animals that had a history of abortion when compared to animals which had no history of abortion (2.4%). Abortion history was a significant potential risk factor (Table 3). The seroprevalence of brucellosis was higher in those animals which were not sharing sires (5.4%) when compared to animals which had their own sire (2.1%). The sharing of sires was found to be a potential risk factor (*p* < 0.05) (Table 3). Interestingly, herds reared on cleaned corals show higher prevalence (14.2%) when compared to those which were reared on uncleaned corals (8.1%), but this finding was statistically insignificant (*p* > 0.05).

A high seroprevalence was observed in healthy animals (7.4%) than in medium (2.3%) and weak (0%) ones. This finding was statistically significant (*p* < 0.05). The seroprevalence regarding stock replacement practice was also determined. Seroprevalence was higher in those herds which purchased replacements (31.5%) as compared to those which reared the replacements on their own (10.0%). This finding was statistically significant (*p* < 0.05). Finally, based on the binary logistic regression analysis, abortion history and the sharing of sires were found to be a potential risk factors (*p* < 0.05) for the seropositivity of animal brucellosis (Table 4).

All seropositive samples (*n* = 35) were also positive in the *Brucella* genus-specific (bcsp31) RT-PCR. Out of the 35 seropositive serum samples, DNA of *Brucella melitensis* was detected in all (100%) samples by species-specific RT-PCR. However, no seropositive serum samples were positive for *B. abortus* species-specific RT-PCR (Table 5).

## 4. Discussion

Brucellosis is a zoonotic infectious bacterial disease, spreading worldwide and affecting animals or humans in developed and developing countries. This disease has negative effects on the economic prosperity of states and has major effects on human health and animal industry [20]. Loss of offspring, temporary or permanent infertility, and reduction in milk production are the adverse effects of brucellosis on livestock production. This disease is found to be more common in developing countries because consumers and farmers of developing countries are unaware of this disease, the possible risk, and appropriate countermeasures [21,22].

In this study, the overall seroprevalence was recorded to be 3.23%, which is higher than in a previous study conducted in the Punjab region of Pakistan, where it was recorded to be 1.6% [15]. However, our findings in this study are lower as compared to the results recorded (8.6%) in the Potohar Plateau, Pakistan [23]. Lower seroprevalence in the present study might be a geographical trend of brucellosis in the Potohar Plateau area.

A considerably higher prevalence of disease was recorded for camels (9.03%) for the Faisalabad district [24]. The possible reason for the seropositivity of other livestock species (i.e., buffalo, cows, goats, and sheep) might be that the same geographical area was included in present study (i.e., Faisalabad).

These results can be compared with these developing countries with cases of brucellosis in livestock. In Kampala, Uganda, 5% of animals were found to be seropositive [25]. A considerably higher number of goats and sheep, i.e., 9.7% and 16%, were found to be seropositive for brucellosis in the Afar and Somali regions of Ethiopia [26]. It can be suspected that poor local herd management might be an important factor for the spread of disease.

The seroprevalence of animal brucellosis was higher in the Faisalabad district (11.3%) than in the other districts (Okara, 4.1%; Lahore, 2.0%; Kasur, 1.3%). A comparatively lower prevalence (5 to 8.5%) was recorded in previous studies conducted in different regions of Pakistan [24,27]. A possible reason for the high seroprevalence of brucellosis in the study area is that livestock holders do not have the ability to find positively tested animals. Most of the livestock farmers are poor and they cannot afford to cull positive animals. So, the mixing of positive animals to healthy herds might spread the disease.

Seroprevalence was higher in buffaloes (5.1%) as compared to cattle (3.8%), sheep (3.4%), and goats (1.8%). A previous study in Nigeria found a higher prevalence of brucellosis in goats (2.83%) [28]. In Egypt, the prevalence was slightly higher in cattle (5.44%), but in buffaloes the prevalence was low (4.11%) [29]. A considerably higher seroprevalence was observed in sheep (5.94%) and goats (6.19%) in Mymensingh, Bangladesh [30]. A possible reason for the observed variations might be the different local herd management systems.

Seroprevalence was higher in male animals (7.4%) than in female animals (2.5%), which was quite the opposite to the situation in Michoacan, Mexico, for small ruminants (male 5%, female 9%) [31]. The data are also in contrast to a previous study conducted in small ruminants in the Potohar Plateau, (Islamabad, Rawat, and Kherimurat) where higher seroprevalence was recorded in females (10.4%) than males (3.03%) [23]. Males are considered less susceptible to infection, due to the absence of erythritol in their reproductive organs [32]. Again, the most plausible reason for the high seroprevalence in males is that farmers do not cull or dispose these animals with brucellosis, where instead they use them for breeding purpose or sell them to other farmers.

Two breeds of goats (Beetle, 1.8%; Teddy, 1.9%) were investigated in this study. Differences in breed prevalences were observed, which is a finding that was observed before in Mexico [31] and in Pakistan [21]. In this study, only one breed of cattle, buffalo, and sheep was observed to be positive to brucellosis (Nili Ravi 5.1%, Rojhan 3.8%, Kajli 3.4%), which was in accordance to earlier findings observed in Pakistan [33]. The possible reason for this result is that the local breed in Pakistan may be more resistant to infection.

This study indicates that sexually mature animals were mainly infected (3.2%). A similar finding was also described previously [34]. Most of the brucellosis infected animals are sexually mature. The multiplication and growth of *Brucella* is stimulated by erythritol and sex hormones, which increase with age and sexual maturity [35].

The prevalence of brucellosis was higher in large herds (24.4%) as compared to small herds (5.2%). Similarly, higher prevalence (56.3%) in large herds was observed in Uganda [25] and in the Potohar Plateau, Pakistan [33]. It is a well-known fact that animals of larger herds have a higher probability of getting into contact with infected animals.

The major symptom of brucellosis in breeding animals is abortion at an advanced stage of pregnancy [36]. In the present study, those animals which have had an abortion history were found to be associated with a higher prevalence of infection (9.0%) than those which had no history of abortion (2.4%). This finding is comparable to the findings conducted in Uganda, [25] Pakistan [33], and Kenya for cattle [37]. The higher rate of infection in aborted animals might be due to no disposal of *Brucella*-seropositive animals, thus enabling infected animals to transmit their infection to other healthy animals.

A higher prevalence of brucellosis was present in those herds with the sharing of sires (5.4%) compared to those with no sharing practice (2.1%). The possible reason of this is that infected males may transfer the disease during mating with females. Interestingly, higher prevalences were shown in animals which were kept in cleaned corals (14.2%) compared to those animals which were kept in uncleaned corals (2.0%). The reasonable explanation of this observation that infected animals were already present in those large herds which were kept in cleaned corals.

An interesting finding of this study is that the prevalence of brucellosis was higher in those animals which were healthy (7.4%) than in medium and weak ones. Our findings are in contrast to the findings of a previous study conducted in Punjab, Pakistan, in which the prevalence of brucellosis was higher in animals which were weak (6.67%) [23]. Also, another study conducted in Ethiopia in small ruminants reported a higher prevalence in weak (6.60%) animals as in medium (0.67%) or healthy ones [38]. The possible reason of controversy between our findings and the previous study might be a poor management system adopted in study area, which could be why a high prevalence was also noticed in healthy livestock.

In this study, stock replacement at the herd level was a potential risk factor of brucellosis. Higher prevalence was recorded in purchased animals (31.5%) than those self-reared (10.0%), which is in contrast to the finding of the prevalence of bovine brucellosis in stock replacement (self-reared 7.6% and purchased 5.9%) in the Potohar region, Pakistan [33]. A comparable result to the present study was observed in other farm animals [38,39]. Controlling the spread of brucellosis is only possible when positive animals are culled and disposed. Brucella-positive trade animals must be prohibited and official resources like reimbursement need to be set in force.

The molecular detection of *Brucella* genus-specific DNA confirmed the results obtained with RBT positive samples of goats, sheep, buffaloes, and cows. In Pakistan and in other countries, Bcsp-31 qRT-PCR and other PCR assays were used to amplify *Brucella* DNA from serum samples [20,23]. With the species-specific PCR assay, *B. melitensis* was confirmed as the causative agent of brucellosis in cattle, sheep, and goats. Similar PCR-based detection of *B. melitensis* from buffaloes, cattle, and goats has been reported from other regions of Pakistan [39,40] and neighboring countries i.e., China, India, and Iran [41,42,43]. An interesting finding is that no evidence for of *B. abortus* was found as the causative agent of brucellosis. However, *B. abortus* was identified as the causative agent of brucellosis in sheep and goats by different authors at the national and international level [21,44]. *B. abortus* was earlier confirmed as the causative agent of brucellosis in sheep, using the bacteriology technique [45].

## 5. Conclusions

In present study, only one species of brucellosis, i.e., *Brucella melitensis*, was identified as the causative agent of animal brucellosis in the Punjab, Pakistan. Diseased animals may act as sources of infection for other animals. The elimination of positive seroreactors, the development of control strategies for brucellosis, and education programs regarding the control of zoonotic disease are highly in need in developing countries like Pakistan.

## Figures and Tables

**Figure 1 microorganisms-07-00449-f001:**
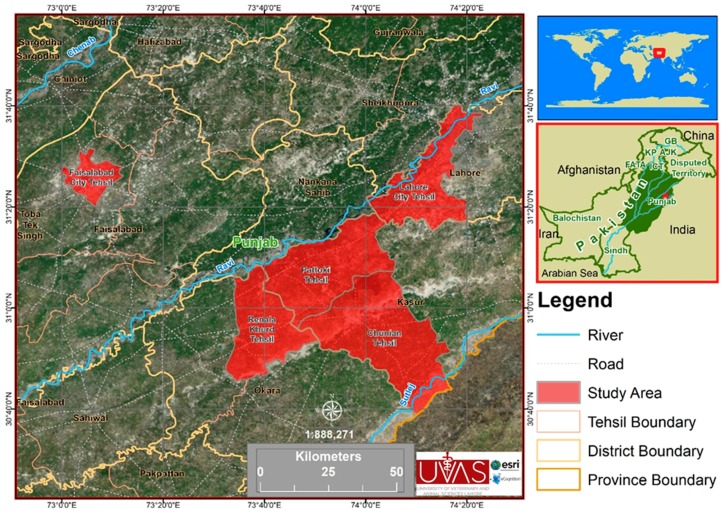
Study area map consisting of five tehsils (Pattoki, Chunian, Renala Khurd, Lahore, and Faisalabad) and four districts (Kasur, Lahore, Okara, and Faisalabad).

**Table 1 microorganisms-07-00449-t001:** Primers and probes for the *Brucella* genus and the species-specific quantitative real-time polymerase chain reaction (qRT-PCR).

Primers and Probes	*Brucella* genus	*Brucella abortus*	*Brucella melitensis*
Forward	5′GCTCGGTTGCCAATATCAATGC3′	5′GCGGCTTTTCTATCACGGTATTC3′	5′AACAAGCGGCACCCCTAAAA3′
Reverse	5′GGGTAAAGCGTCGCCAGAAG3′	5′CATGCGCTATGATCTGGTTACG3′	5′CATGCGCTATGATCTGGTTACG3′
Probe	5′AAATCTTCCACCTTGCCCCTTGCCATCA3′	5′CGCTCATGCTCGCCAGACTTCAATG3′	5′CAGGAGTGTTTCGGCTCAGAATAATCCACA3′
5′Flourophore/3′ Quencher	6-Fam/BHQ_1	HEX/BHQ_1	Cy5/BHQ_2
Target Gene	bcsp31	IS711 element downstream of the alkB	IS711 element downstream of BMEI1162

**Table 2 microorganisms-07-00449-t002:** District and tehsil-wise seroprevalence of brucellosis.

Variables	Categories	Sample Examined	Sample Positive (%)	Statistical Analysis	
				Chi Square Value	*P*-Value
District	Faisalabad	150	17 (11.3)	39.947	0.000
	Okara	169	7 (4.1)		
	Lahore	152	3 (2.0)		
	Kasur	612	8 (1.3)		
Tehsil	Chunian	12	2 (16.7)	49.181	0.000
	Faisalabad	150	17 (11.3)		
	Renala Khurd	169	7 (4.1)		
	Lahore	152	3 (2.0)		
	Pattoki	600	6(1.0)		

**Table 3 microorganisms-07-00449-t003:** Risk factors associated with ovine, caprine, and bovine brucellosis in the four districts of Punjab, Pakistan, on the basis of Chi square analysis.

Variables	Factors	Sample Examined	Positive for RBPT (%)	Statistical Analysis	
				Chi Square Value	*P*-Value
Species	Buffalo	234	12 (5.1)	5.813	0.121
	Cow	206	8 (3.8)		
	Sheep	203	7 (3.4)		
	Goat	440	8 (1.8)		
Gender	Male	148	11 (7.4)	9.673	0.000
	Female	935	24 (2.5)		
Breed	Nili Ravi	234	12 (5.1)	5.815	0.213
	Rojhan	206	8 (3.8)		
	Kajli	203	7 (3.4)		
	Teddy	52	1 (1.9)		
	Beetle	388	7 (1.8)		
Age	Adult	1065	35 (3.3)	0.611	0.434
	Young	18	0 (0)		
Herd size (*n* = 149)	≤10	96	5 (5.2)	13.803	0.000
	>10	53	14 (26.4)		
Abortion history	No	950	23 (2.4)	16.258	0.000
	Yes	133	12 (9.0)		
Have their own Sire	No	368	20 (5.4)	8.650	0.000
	Yes	715	15 (2.1)		
Cleaning up the coral in herd (*n* = 149)	No	37	3 (8.1)	0.954	0.329
	Yes	112	16 (14.2)		
Body condition	Weak	20	0 (0)	14.304	0.001
	Medium	861	20 (2.3)		
	Healthy	202	15 (7.4)		
Stock replacement in herd (*n* = 149)	Self-reared	130	13 (10.0)	6.938	0.008
	Purchased	19	6 (31.5)		

**Table 4 microorganisms-07-00449-t004:** Final model with associated risk factors for animal brucellosis-binary logistic regression analysis.

Variables	Binary Logistic Regression			*P*-Value
	OR	(95% CI)		
		Lower	Upper	
District	1.484	0.987	2.231	0.058
Tehsil	1.407	0.976	2.029	0.068
Herd Size	0.639	0.147	2.771	0.550
Species	0.871	0.518	1.465	0.604
Gender	0.918	0.286	2.954	0.886
Breed	1.113	0.788	1.574	0.543
Age	21763518.14	0.000		0.999
Abortion history	4.603	1.853	11.430	**0.001**
Have their own sire	0.230	0.086	0.612	**0.003**
Cleaning up the coral	0.469	0.094	2.338	0.356
Body condition	1.440	0.950	2.181	0.085
Stock replacement	1.542	0.309	7.685	0.597

OR: Odds ratio.

**Table 5 microorganisms-07-00449-t005:** Seroprevalence and molecular detection of *Brucella* DNA in tested serum samples.

Case No.	Host	Serological Assay	PCR		
		RBPT	*Brucella* Genus	*B. abortus*	*B. melitensis*
94	Goat	+	+	−	+
160	Goat	+	+	−	+
167	Goat	+	+	−	+
192	Goat	+	+	−	+
198	Sheep	+	+	−	+
278	Goat	+	+	−	+
309	Goat	+	+	−	+
315	Goat	+	+	−	+
316	Goat	+	+	−	+
319	Sheep	+	+	−	+
320	Sheep	+	+	−	+
322	Sheep	+	+	−	+
324	Buffalo	+	+	−	+
334	Buffalo	+	+	−	+
342	Cow	+	+	−	+
348	Cow	+	+	−	+
350	Cow	+	+	−	+
440	Sheep	+	+	−	+
456	Buffalo	+	+	−	+
463	Buffalo	+	+	−	+
470	Buffalo	+	+	−	+
473	Buffalo	+	+	−	+
479	Cow	+	+	−	+
489	Cow	+	+	−	+
491	Cow	+	+	−	+
499	Cow	+	+	−	+
503	Buffalo	+	+	−	+
504	Buffalo	+	+	−	+
506	Buffalo	+	+	−	+
513	Buffalo	+	+	−	+
517	Buffalo	+	+	−	+
553	Buffalo	+	+	−	+
569	Cow	+	+	−	+
578	Sheep	+	+	−	+
580	Sheep	+	+	−	+

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
