# Peer review of "Seroepidemiology and the Molecular Detection of Animal Brucellosis in Punjab, Pakistan"

_microorganisms, 2019, doi:10.3390/microorganisms7100449_

Round 1

Reviewer 1 Report

Revision of Manuscript ID: microorganisms-594780

Dear Authors,

Your manuscript entitled “Seroepidemiology and molecular detection of animal brucellosis in Punjab, Pakistan” reports an interesting survey on Brucellosis in different domestic animal species in a area of Pakistan. Epidemiological investigations are of great interest and utility. The data reported are interesting. The study is well planned and conducted

However, I have some questions and some modifications are required:

Introduction Lines 48-53: probably this part is unnecessary; Line 54: “The main symptoms…” in my opinion it is better to specify that Authors are speaking of animals; Material and Methods Line 90: it would be better to list the provinces in the same order they were described in the text subsequently; Lines 91-92: “… is famous as it is the burial place of great Sufi poet Bulleh Shah…” it is an interesting information, but is it necessary? Lines 93-95: for the other Districts, Authors reported data from the “9211 system”; where did they collected the data in this case? Lines 95-97, 108-110, 122-123: Could the Authors provide bibliography to support these statements? Lines 111-117: Authors should report data about disease and vaccination for Lahore as for the other districts; Lines107, 113, 120: The Authors reported the system in 2 different way: “9211 system”, “9211 system-livestock Punjab”; Line 117: “Faisalabad (301.4504°N, 73.1350°E) is the third largest city of the province Punjab.” This sentence should be linked to the subsequently paragraph; Lines 142-148: Authors should call the test in the same way, RBT or RBPT. Lines 158-159: “Primer and probes that used were supplied by TIB MOLBIOL (Berlin, Germany) (Table 1).” This sentence should probably be reformulated, like for example “Used primers and probes were supplied ….” Results Lines 187-188: this sentence should probably be reformulated; Line 191: Probably “aborted animals” is not the best definition; it would be better to call them “animals that had/gave aborted” or “animals that had history/anamnesis of abortion”; Table 3: It would be better to divide the table making one table with the result of Animals and one with the results of herds; Table 5: What do the Authors mean with “NA” for melitens PCR of sample 198? Explanation of “NA” must be reported in Table legend and some information should be given in the text. Discussion Lines 248-250: this conclusion is not appropriate; The cited work speak about a different animal species in a different geographic area; the only common District is Faisalabad that presented a seroprevalence (9.03%), in camels, very similar to the percentage of positivity recorded by the Authors in the same District; Your conclusion is proper, but only for the Faisalabad District; for this reasons the sentence should be reformulated; Lines 256-263: this paragraph should be reformulated for the same reasons reported above; Lines 275-277: this is a very interesting consideration. Could the Authors provide information about the average length of life of male and female animals to support this? Lines 296-297: in my opinion, this statement is not exact; seropositivity is more often associated with abortion because abortion is one of the main symptoms of brucellosis in farm animals and it is not linked to insemination method; artificial insemination could prevent spreading and venereal transmission, but not abortion directly. Lines 300-313: Did the Authors considered the possibility that the higher positivity found in animals kept in cleaned corals and healthy could be linked to the herd size and it could relate to these animals come from the biggest farms? Line 321-322: in my opinion this sentence should reformulated, like, for example: “Molecular detection of Brucella genus-specific DNA confirmed the results obtained with RBPT.” Lines 325-328: The Authors put the attention on the absence of abortus (and this could be eliminated or reduced), but they did not discus the presence of B. melitensis in the investigated samples, especially the ones collected from cows and buffalos.

Author Response

Point1: Introduction Lines 48-53: probably this part is unnecessary;

Response: We thank the reviewer for her/his comment.

We removed the paragraph

Point 2: Line 54: “The main symptoms…” in my opinion it is better to specify that Authors are speaking of animals;

Response 2: Line 48 [now] revised as per suggestion “The main symptoms of brucellosis in animals are abortion, reduction of milk production, hygroma, neonatal mortality and infertility in females while, in male are orchitis and epididymitis [5].

Point3: Material and Methods Line 90: it would be better to list the provinces in the same order they were described in the text subsequently

Response 3: revised as per suggestion ‘The study was conducted in Kasur, Okara, Lahore and Faisalabad’ [now line 84]

Point 4: Lines 91-92: “… is famous as it is the burial place of great Sufi poet Bulleh Shah…” it is an interesting information, but is it necessary?

Response 4: Line 85-86 [now] are revised “District Kasur (31.1165° N, 74.4494° E) borderes in the South to Lahore and India, the Ganda Singh border.”

Point 5: Lines 93-95: for the other Districts, Authors reported data from the “9211 system”; where did they collected the data in this case?

Response 5: we collected data for District Kasur from same source “9211 system” as for other districts. Line Line 87-88 [now] revised “According to 9211 virtual governance system livestock of this district includes large animals (1,060,994), poultry (243,141) and small ruminants (319,048).

Point 6: Lines 95-97, 108-110, 122-123: Could the Authors provide bibliography to support these statements?

Response 6:  Bibliography included as per suggestion NOW Line 90, 104, 113, 376-377.

Point 7: Lines 111-117: Authors should report data about disease and vaccination for Lahore as for the other districts

Response 7: Disease and vaccination data included now “The prevalent diseases of this area related to livestock are milk fever, mastitis, herpes simplex virus, bovine ephemeral fever, Foot and Mouth disease. Vaccinations are applied [16].” Line 111-113 NOW, LINE 104,

Point 8: Lines107, 113, 120: The Authors reported the system in 2 different way: “9211 system”, “9211 system-livestock Punjab”

Response 8: correct system cited now as “9211 virtual governance system” NOW Line 87, 101, 107-108, 116.

Point 9: Line 117: “Faisalabad (301.4504°N, 73.1350°E) is the third largest city of the province Punjab.” This sentence should be linked to the subsequently paragraph;

Response 9: Done as per suggestion Line 114-115 Now

Point 10: Lines 142-148: Authors should call the test in the same way, RBT or RBPT.

Response 10: we are using RBPT now all over the text, NOW 141 and 143

Point 11: Lines 158-159: “Primer and probes that used were supplied by TIB MOLBIOL (Berlin, Germany) (Table 1).” This sentence should probably be reformulated, like for example “Used primers and probes were supplied ….”

Response 11. Sentence revised as per suggestion “Used primers and probes were supplied by TIB MOLBIOL (Berlin, Germany)” Line 154-155 now

Point 12: Results Lines 187-188: this sentence should probably be reformulated;

Response 12: sentence reformulated “The Brucella antibodies were detected in 19 (12.75%) herds which classified into 6 herds goat, 5 herds buffalo, 4 herds cow and 4 herds sheep..” Line 185-186 now

Point 13: Line 191: Probably “aborted animals” is not the best definition; it would be better to call them “animals that had/gave aborted” or “animals that had history/anamnesis of abortion”

Response 13: sentence revised as per suggestions “animals that had history of abortion” Line 189 now

Point 14: Table 3: It would be better to divide the table making one table with the result of Animals and one with the results of herds

Response 14: confusion of results for animals and herds has been removed now, by adding herd number (n=149) along with parameters like Stock Replacement in herd (n=149)

Point 15: Table 5: What do the Authors mean with “NA” for melitens PCR of sample 198? Explanation of “NA” must be reported in Table legend and some information should be given in the text.

Response 15: this sample was also positive for B. melitensis. Corrected now in table

Point 16: Discussion Lines 248-250: this conclusion is not appropriate; The cited work speak about a different animal species in a different geographic area; the only common District is Faisalabad that presented a seroprevalence (9.03%), in camels, very similar to the percentage of positivity recorded by the Authors in the same District; Your conclusion is proper, but only for the Faisalabad District; for this reasons the sentence should be reformulated

Response 16: sentence revised “A considerably higher prevalence of disease was recorded for camels (9.03%) for the district Faisalabad [22]. The possible reason for the seropositivity of other livestock species (i.e. buffalo, cow, goat, and sheep) might be that the same geographical area was included in present study (i.e. Faisalabad). Now line 246-249.

Point 17: Lines 256-263: this paragraph should be reformulated for the same reasons reported above

Response 17: paragraph reformulated, now line 255 to 260

Point 18: Lines 275-277: this is a very interesting consideration. Could the Authors provide information about the average length of life of male and female animals to support this?

Response 18: The average length of life of goat has an expectancy of around 15 to 20 years. The average lifespan of does is around 17 years, whereas males can die as young as 8 years old. Does may live shorter lives if they experience difficult births. Bucks don’t live as long as does because of the stress of the rut. The life expectancy of a sheep is 10 to 12 years, though some sheep may live as long as 20 years. However, the length of a sheep's productive lifetime tends to be much less. This is because a ewe's productivity usually peaks between 3 and 6 years of age and begins to decline after the age of 7. As a result, most ewes are removed from a flock before they would reach their natural life expectancy. Depends on the type or class, modern dairy cattle have an average lifespan between 5 to 6 years. Cattle raised for beef have a lifespan between 12 to 24 months. Beef cattle for breeding have an average lifespan of around 8 to 12 years. Cattle have been known to live over twice as long as what's considered average, depending on breed, productivity, health status and temperament. Water buffalo tend to live around 25 years, while African buffalo live around 26 years.

Point 19: Lines 296-297: in my opinion, this statement is not exact; seropositivity is more often associated with abortion because abortion is one of the main symptoms of brucellosis in farm animals and it is not linked to insemination method; artificial insemination could prevent spreading and venereal transmission, but not abortion directly.

Response 19: statement revised “The higher rate of infection in aborted animals might be due to no disposal of Brucella-seropositive animals thus enabling infected animals to transmit infection to other healthy animals” now Line 293-295

Point 20: Lines 300-313: Did the Authors considered the possibility that the higher positivity found in animals kept in cleaned corals and healthy could be linked to the herd size and it could relate to these animals come from the biggest farms?

Response 20: changed

Point 21: Line 321-322: in my opinion this sentence should reformulated, like, for example: “Molecular detection of Brucella genus-specific DNA confirmed the results obtained with RBPT.”

Response 21: sentence revised, now Line 317

Point 22: Lines 325-328: The Authors put the attention on the absence of abortus (and this could be eliminated or reduced), but they did not discus the presence of B. melitensis in the investigated samples, especially the ones collected from cows and buffalos.

Response 21: discussion related to B. melitensis added, now line 321-323

Reviewer 2 Report

The manuscript ID: microorganisms-594780 entitled ‘Seroepidemiology and molecular detection of animal brucellosis in Punjab, Pakistan’ deals with studies, which aimed to determine the prevalence of the genus Brucella affecting domestic animals in Pakistan. The authors have done important work investigating the quantification of Brucella in domestic animal with supporting their presence. The writing is good and the data are presented properly in a clear form.

However, I have one concern in this manuscript. The authors need to sequence and deposit the obtained nucleotide sequences in the GenBank database and if available, please add some phylogenetic analysis in the manuscript to enforce the robustness of their investigation.

Additionally, authors should more discuss on the other Brucella study in Pakistan and near countries as the pathogen can be transmitted through free-ranging wild animals. Please add some discussions and proper references in the manuscript.

Author Response

Point1: However, I have one concern in this manuscript. The authors need to sequence and deposit the obtained nucleotide sequences in the GenBank database and if available, please add some phylogenetic analysis in the manuscript to enforce the robustness of their investigation

Response 1: In this study, we aimed to identify the Brucella DNA in serum samples collected from different animal species using real time PCR. The concentration of DNA extracted from serum is not sufficient to apply sequencing of them.

Point 2: Additionally, authors should more discuss on the other Brucella study in Pakistan and near countries as the pathogen can be transmitted through free-ranging wild animals. Please add some discussions and proper references in the manuscript.

Response 2: possible discussion added along with reference, Line 321 to 323 and 439 to 446.

Round 2

Reviewer 1 Report

Dear Authors,

I appreciate the corrections you made, following the suggested modifications.

Author Response

Point1:

 I appreciate the corrections you made, following the suggested modifications.

Response:

We are grateful to reviewer for his/her positive response

Reviewer 2 Report

Among the two questions, the second one was revised properly.

However, in my opinion, the authors must confirm their positive amplicons from qPCR by using sequencing analysis or at least describing the information regarding the positive control during the qPCR analysis.

If it it not available to analyze the exact sequence from the entire positive amplicons, please describe the detailed information of positive control and its use during qPCR analyses.

Author Response

Point1:

Among the two questions, the second one was revised properly.

However, in my opinion, the authors must confirm their positive amplicons from qPCR by using sequencing analysis or at least describing the information regarding the positive control during the qPCR analysis.

If it it not available to analyze the exact sequence from the entire positive amplicons, please describe the detailed information of positive control and its use during qPCR analyses.

Response 1:

The information about positive and negative controls and their use is described now “The PCR reaction and analysis were performed using Mx3000P Thermocycler (Stratagene, Canada). The samples scored positive by the instrument were additionally confirmed by visual inspection of the graphical plots showing cycle numbers versus fluorescence values [19]. Reference strain of B. abortus S-99 (ATCC 23448) and B. melitensis 16M (ATCC 23456) were provided from NRL-Brucellosis as positive control. A non-Brucella gram-negative strain used in the present study to evaluate the specificity of the primers and real time PCR reaction was E. coli (ATCC 10538). A sample with a fluorescence signal 30 times greater than the mean standard deviation in all wells over cycles 2 through 10 was considered a positive result, whereas a sample yielding a fluorescence signal less than this threshold value was considered negative. Cycle threshold values below 38 cycles were interpreted as positive. The threshold was set automatically by the instrument. The samples scored positive by the instrument were additionally confirmed by visual inspection of the graphical plots showing cycle numbers versus fluorescence values”. Line 156-167